# Effect of MWCNT Anchoring to Para-Aramid Fiber Surface on the Thermal, Mechanical, and Impact Properties of Para-Aramid Fabric-Reinforced Vinyl Ester Composites

**Jinsil Cheon [1,2] and Donghwan Cho [1,*]**

1   Department of Polymer Science and Engineering, Kumoh National Institute of Technology, Gumi 39177, Gyeong-buk, Republic of Korea; jscheon@kotmi.re.kr

2   Composites Convergence Research Center, Korea Textile Machinery Convergence Research Institute, Gyeongsan 38542, Gyeong-buk, Republic of Korea

*   Correspondence: dcho@kumoh.ac.kr

**Abstract:** In the present work, para-aramid fabrics (p-AF) were physically modified via an anchoring process of 0.05 wt% MWCNT to the aramid fiber surfaces by coating the MWCNT/phenolic/methanol mixture on p-AF, and then by thermally curing phenolic resin of 0.01 wt%. Para-aramid fabric-reinforced vinyl ester (p-AF/VE) composites were fabricated using p-AF/VE prepregs by compression molding. The effect of MWCNT anchoring on the thermo-dimensional, thermal deflection resistant, dynamic mechanical, mechanical, and impact properties and the energy absorption behavior of p-AF/VE composites was extensively investigated in terms of coefficient of linear thermal expansion, heat deflection temperature, storage modulus, tan δ, tensile, flexural, and Izod impact properties and a drop-weight impact response. The results well agreed with each other, supporting the improved properties of p-AF/VE composites, which were attributed to the effect of MWCNT anchoring performed on the aramid fiber surfaces.

**Keywords:** para-aramid fiber; carbon nanotube; anchoring; modification; vinyl ester; composite; properties

## 1. Introduction

In the field of soft body armors, research challenges have included reducing weight, improving flexibility, and enhancing impact resistance, simultaneously [1–3]. Para-aramid (referred to as p-aramid hereinafter) fiber has been frequently used to protect human beings from physical risks under dangerous military and civilian circumstances, because polymeric chains consisting of the fiber form rigid crystals and are aligned with the fiber direction during fiber spinning [3,4]. Substantially, the surface of p-aramid fiber contains many hydrophilic functional groups, such as amide and hydroxyl groups, which can form a huge number of hydrogen bonds between the polymer chains. Owing to the intrinsic structure of p-aramid fiber, it can resist severe-impact environments without fiber breakages [5].

For this reason, woven fabrics made with p-aramid fibers have been used for protective clothing. According to the energy-dissipating mechanism, the friction between the inter-yarns in the woven fabric with a plain pattern plays a significant role in the ballistic impact response in both direct and indirect manners [6–10]. A direct effect of the inter-yarn friction is that the energy dissipation is increased when the yarns consisting of the fabric begin to displace one another, exhibiting sliding, pulling-out, or re-orientation of the individual fibers. An indirect effect of the inter-yarn friction is that it may influence external forces, which can be transferred and redistributed to the neighboring yarns [11].

P-aramid fabric-reinforced composites have been considered as key materials in many civilian and military applications due to their excellent specific strength, specific stiffness, and lightness in comparison to conventional materials, such as metals and alloys. They

exhibit excellent impact resistance and elasticity because of the combination of the viscoelasticity of the polymer matrix and the impact toughness of p-aramid fabrics consisting of the composites [12].

In general, soft armor made with p-aramid fabric requires weak fiber–matrix interfacial adhesion because the friction between the fabric layers critically influences the anti-bulletproofing performance. However, when p-aramid fiber-reinforced polymer composites are used for ballistic protection in the plate form, as in hard armors, the strong interfacial bonding between the fiber and the polymer matrix is important because the composite may experience maximum deformation and can absorb the highest energy upon impact. Therefore, in the case of hard armors, thermosetting polymer matrices have advantages over thermoplastic polymer matrices with low stiffness and high deformation [4].

For the past years, many experimental and theoretical studies have been carried out to understand the material's response and the penetration failure mechanism as well as the energy absorption upon ballistic impact [12–18]. Pandya et al. [17] and Sarasini et al. [18] reported on the ballistic impact behavior of hybrid epoxy composites reinforced with basalt and p-aramid fabrics for hard body armors. Each fabric was laminated with an alternating sequence to investigate the effect of each fabric layer on the composite performance in terms of impact energy absorption capability and enhanced damage tolerance. Davidovitz et al. [19] studied the failure mode and fracture mechanism of Kevlar/epoxy composites under flexural deformation. The failure mode was described in terms of tensile failure and delamination. The tensile failure of the composite was explained by fiber splitting, fiber pull-out, delamination, fiber bending, tearing-off of the fiber skin, and shearing of the individual filaments. Wang et al. [20] studied the crushing behaviors and mechanisms of composite thin-walled structures under quasi-static compression and dynamic impact conditions. They addressed that fiber-reinforced composite structures and materials showed good potential for solving impact problems and energy absorption.

It has been well known that multi-walled carbon nanotubes (MWCNT), which exhibit a high aspect ratio and a large specific volume, are a promising material to improve the mechanical, thermal, electrical, and tribological properties of polymer composites [20–22]. It has been emphasized that a key factor to introduce MWCNT to the polymer matrix is good dispersion. Therefore, many papers have been studied on surface modification of MWCNT to enhance the dispersity and the interfacial bonding between the MWCNT and the polymer matrix, frequently focusing on the chemical functionalization of MWCNT [23–29]. However, chemical functionalization or grafting often requires complicated procedures and large quantities of chemicals.

One of the simplest experimental approaches to incorporate MWCNT into a fiber-reinforced polymer composite material is anchoring [30,31]. Here, the word 'anchoring' refers to a process physically attaching MWCNT nanoparticles on the individual fiber surfaces with the assistance of diluted thermosetting resin at low concentration such that the anchoring process may influence the inter-yarn friction during the pulling-out test of individual yarns. In the case of MWCNT anchoring at high resin concentrations, the fabric drapeability might be lowered to some extent.

Consequently, the objectives of the present work are to physically attach MWCNT to the p-aramid fiber surface by the anchoring process with the assistance of diluted phenolic resin, to fabricate vinyl ester composites reinforced with MWCNT-anchored p-aramid fabrics by a compression molding technique, and finally, to extensively investigate the effect of MWCNT anchoring on the thermo-dimensional, dynamic mechanical, heat deflection temperature, mechanical, and impact properties and the energy absorption behavior of p-aramid fabric-reinforced vinyl ester composites.

## 2. Materials and Methods

### 2.1. Materials

P-aramid fabrics (HERACRON®, HT840, Kolon Industries Co., Ltd., Gumi, Republic of Korea) with a plain weave pattern (referred to as p-AF hereinafter) were used as rein-

forcement in this work. Each fiber yarn has 840 deniers in the warp and weft directions, respectively. The fabric density is 26.7 counts per inch. The areal density is 200 g/cm$^2$. The commercial p-AF was used 'as-received' without further cleaning and surface treatment. MWCNT (CVD-CM95, Hanhwa Chemical Co., Ltd., Seoul, Republic of Korea) were used 'as-received' without further purification and surface treatment. Resole-type phenolic resin (KRD-HM2, Kolon Industries, Co., Ltd., Gimcheon, Republic of Korea) was used as anchoring agent after being diluted with methanol. The phenolic resin contains the solid contents of about 60%. Methanol (99.95% purity, Daejung Chemicals and Metals, Co., Ltd., Siheung-si, Republic of Korea) was used as diluent of phenolic resin.

### 2.2. Preparation of MWCNT/Phenolic/Methanol Mixture and MWCNT Anchoring Process

First, prior to preparation of MWCNT-anchored p-AF, MWCNT nanoparticles were well dispersed in methanol. A mixture of MWCNT/phenolic/methanol was prepared by sufficiently mixing with a magnetic stirrer. The MWCNT in the mixture were uniformly dispersed with phenolic resin by ultrasonication. The ultrasonic process was carried out with the frequency of 40 kHz at 50~60 °C for 1 h using an ultrasonic bath (Model Power Sonic 420, 600 W, Hwashin Co., Ltd., Seoul, Republic of Korea). The concentrations of MWCNT and phenolic resin present in the MWCNT/phenolic/methanol mixture were 0.05 wt% and 0.01 wt%, respectively. Phenolic resin of 0.01 wt% was used because it was optimal to have the appropriate fabric drapeability and to cure it with the MWCNT anchored to the fiber surface, as found earlier [29,30]. Accordingly, anchoring process of MWCNT to p-AF was performed with 0.05 wt% MWCNT and 0.01 wt% phenolic resin.

In this work, anchoring of MWCNT to p-AF by using 0.01 wt% phenolic resin means that MWCNT nanoparticles were physically attached on the surface of individual p-aramid fibers by diluted phenolic resin. For MWCNT anchoring, p-AF was immersed in the MWCNT/phenolic/methanol mixture. At this time, the individual fibers in the fabric were surrounded by the MWCNT dispersed in diluted phenolic resin and the MWCNT nanoparticles were physically attached on the fiber surface of p-AF by curing the diluted phenolic resin at 80 °C for 10 min in a convection oven, as shown in Figure 1.

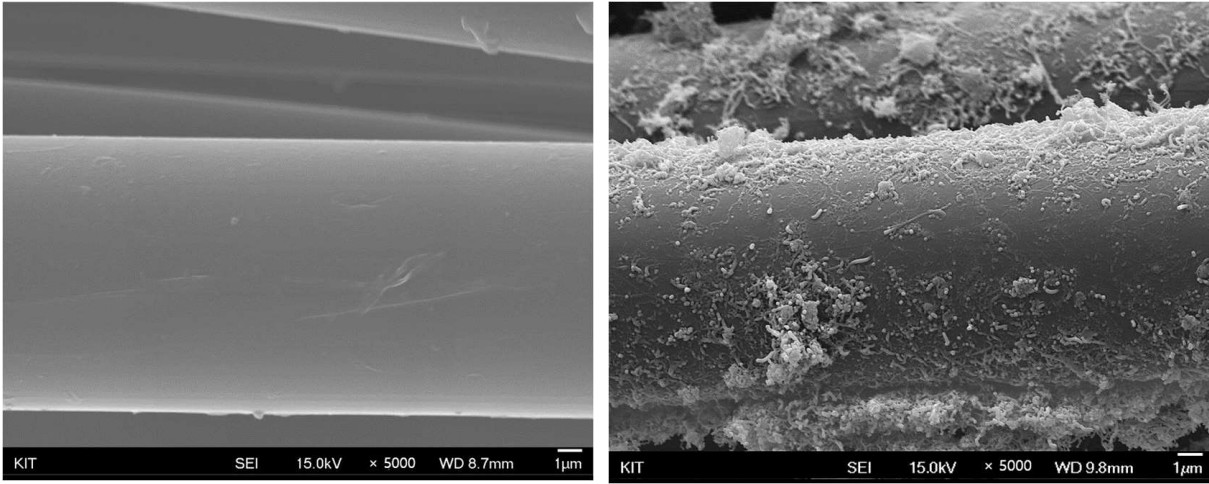

**Figure 1.** Topography (×5000) showing the pristine p-aramid fiber surface without MWCNT (**left**) and the p-aramid fiber surface with MWCNT anchored by thermally curing dilute phenolic resin (**right**).

### 2.3. Fabrication of MWCNT-Anchored P-Aramid Fabric/Vinyl Ester Composites

Bisphenol-A modified epoxy acrylate-type vinyl ester resin (Model RF-1001, CCP Composites Korea Co., Ltd., Wanju-gun, Republic of Korea) (referred to as VE hereinafter) was used as a matrix of composites. It contains 45~55 wt% styrene acting as both reactive diluent and curing agent of VE. The resin density is 1.03~1.11 g/cm$^3$ at 25 °C and the

resin viscosity is 250~450 cP. In this work, p-AF-reinforced VE (referred to as p-AF/VE hereinafter) composites were fabricated by a compression molding process.

Two types of peroxides with different molecular sizes were used together as the initiator to cure VE. One was di(4-tert-butylcyclohexyl)peroxydicarbonate (DPDC) and the other was tert-butyl-peroxybenzoate (TBPB). Figure 2 shows the chemical structures of VE, DPDC, and TBPB. The effect of the dual initiators at various concentrations on the VE curing behavior was extensively studied in our earlier report [32]. It was found that 1 pph (parts per hundred) DPDC and 0.75 pph TBPB were optimal to cure VE in the presence of MWCNT. Accordingly, DPDC of 1 pph and TBPB of 0.75 pph were used in the present work.

**Figure 2.** Chemical structures of (**a**) vinyl ester resin, (**b**) *di*-*tert*-butylcyclohexyl)peroxydicarbonate (DPDC), and (**c**) *tert*-butyl-peroxybenzoate (TBPB).

Prior to composite fabrication, p-AF/VE prepregs were prepared. Each prepreg contained excess VE because part of the VE impregnated in p-AF could be squeezed out by the applied pressure upon compression molding. Each prepreg was partially cured at 70 °C for 10 min in a convection oven for B-staging. P-AF/VE prepregs with and without MWCNT anchoring were also prepared for comparison. To prevent possible unintended curing prior to uses, the prepregs were completely sealed and kept in a freezer.

The p-AF/VE prepregs of 14 plies were regularly stacked in a stainless-steel mold and processed using a compression molding machine (GE-122S, Kukje Scien, Daejeon, Republic of Korea). Figure 3 depicts the experimental procedure to prepare p-AF/VE prepregs and to fabricate the composite via prepreg stacking and compression molding. The stacked prepregs in the mold were heated up to 180 °C with the heating rate of 6 °C/min. A pressure of 6.89 MPa was applied from 40 °C. When the mold temperature reached 70 °C, the debulking step was conducted to degas the entrapped air between the prepregs and to evaporate organic volatiles therein. The debulking step was repeated twice until the mold temperature reached 110 °C. The final curing was performed at 180 °C for 10 min. The applied pressure of 6.89 MPa was maintained until the end of compression molding.

**Step 1: Pre-impregnation process to prepare of p-AF/VE prepreg**

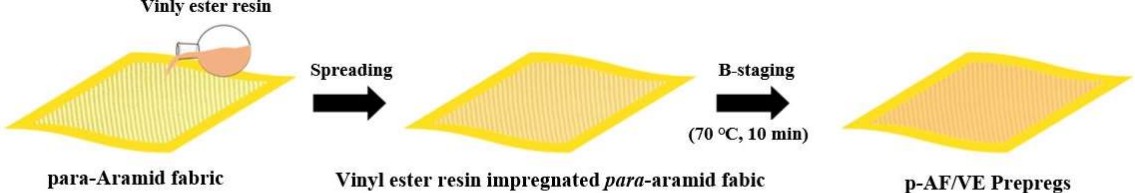

**Step 2: Stacking-up of prepregs and charging in the mold**

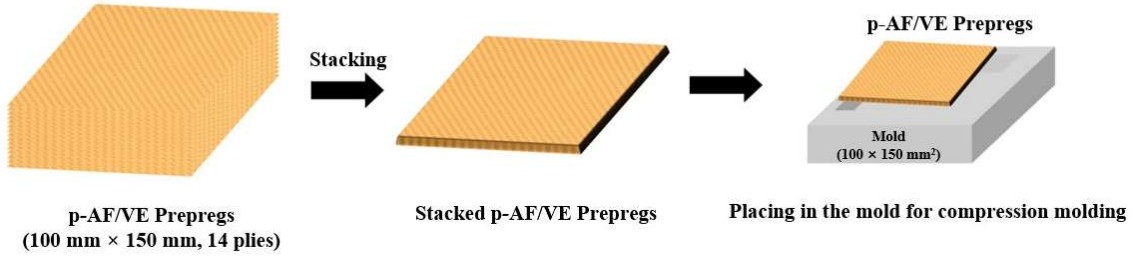

**Step 3: Fabrication of p-AF/VE composite by compression molding process**

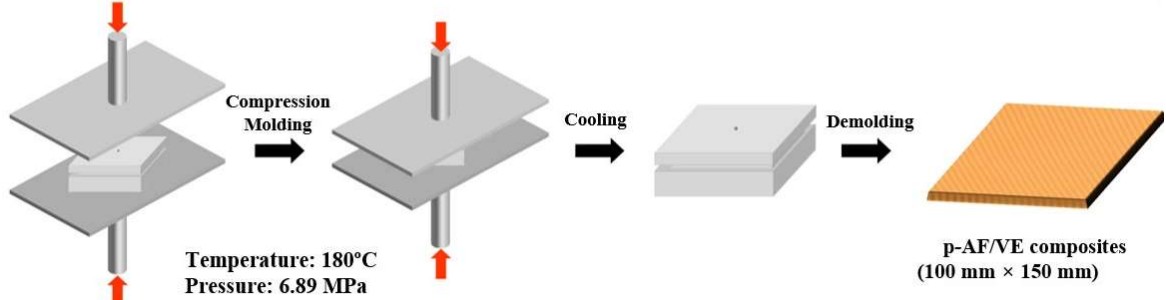

**Figure 3.** Preparation of p-AF/VE prepregs and fabrication of p-AF/VE composite by compression molding.

The molded composite was cooled down to ambient temperature, and then demolded. Finally, p-AF/VE composites with the dimensions of 150 mm × 100 mm × 3 mm were obtained. P-AF/VE composites with and without MWCNT anchoring were prepared for comparison. The p-AF/VE composite with MWCNT anchoring was designated as MWCNT-p-AF/VE composite. The p-AF/VE composite without MWCNT anchoring was designated as pristine p-AF/VE composite.

### 2.4. Microscopic Observation

A field-emission scanning electron microscope (JSM-6500F, JEOL, Tokyo, Japan) was used to observe the topography of MWCNT anchored on the fiber surfaces. Prior to SEM observations, each sample was uniformly coated with platinum for 3 min by a sputtering method. The acceleration voltage was 15 kV, and the secondary electron image (SEI) mode was used.

### 2.5. Thermal Analysis

Thermomechanical analysis (TMA 2940, TA Instruments, New Castle, DE, USA) was performed to investigate the effect of MWCNT anchoring on the thermo-dimensional

stability of p-AF/VE composites. A load of 0.05 N was applied on the specimen using a macro-expansion probe. The thermo-dimensional change was recorded from 30 to 250 °C with the heating rate of 5 °C/min, purging nitrogen gas (50 mL/min).

The effect of MWCNT anchoring on the dynamic mechanical properties of p-AF/VE composites was examined by dynamic mechanical analysis (DMA Q800, TA Instruments, New Castle, DE, USA). The analysis was performed from 30 to 250 °C with the heating rate of 3 °C/min in ambient atmosphere. The dual cantilever mode with a drive clamp and a fixed clamp was used throughout DMA measurement. The oscillation amplitude was 10 μm and the frequency was 1 Hz. The dimensions of composite specimen were 63.5 mm × 12.5 mm × 3 mm.

The heat deflection temperature (HDT) of p-AF/VE composites was measured with a three-point bending mode according to the ASTM D648 standard by using a heat deflection temperature tester (Model 603, Tinius Olsen, Horsham, PA, USA). The dimensions of composite specimen were 127 mm × 12.5 mm × 3 mm. The measurement was performed until the specimen was deflected by 0.254 mm under the bending load of 1.82 MPa. The heating rate of 2 °C/min was used to heat the composite specimen immersed in a silicone oil bath.

### 2.6. Mechanical Test

A universal testing machine (UTM, AG-50kNX, SHIMADZU, Kyoto, Japan) was used to investigate the effect of MWCNT anchoring on the flexural and tensile properties of p-AF/VE composites. Each specimen was cut to fit the mechanical test requirement by using a low-speed diamond saw. The average values of the flexural and tensile properties were obtained from 10 specimens of each composite.

A three-point flexural test was performed according to the ASTM D790 standard. The dimensions of composite specimen were 127 mm × 12.5 mm × 3 mm. The span-to-depth ratio was 32:1. The load cell of 50 kN and the crosshead speed of 5 mm/min were used. A tensile test was performed according to the ASTM D3039 standard. The dimensions of composite specimen were 140 mm × 12.5 mm × 3 mm. The gage length was 80 mm. The load cell of 50 kN and the crosshead speed of 5 mm/min were used. The flexural and tensile tests were repeated 10 times for each sample. The average values of the flexural modulus, flexural strength, tensile modulus, and tensile strength were obtained from 10 repetitive tests of each composite, respectively.

### 2.7. Izod Impact Test and Weight-Drop Impact Test

The Izod impact test was carried out according to the ASTM D256 standard using an impact test machine (IT 892, Tinius Olsen, Horsham, PA, USA). The dimensions of composite specimen without notch were 63.5 mm × 12.5 mm × 3 mm. A pendulum energy of 3.17 J was used. The average value of impact strength was obtained from 10 specimens of each composite.

To inspect the effect of MWCNT anchoring on the energy absorption behavior of p-AF/VE composites exposed to high-speed impact energy, drop-weight impact test (CEAST 9350, Instron, Norwood, MA, USA) was performed according to the ASTM D5628 standard. The dimensions of composite specimen were 100 mm × 100 mm × 4 mm. The diameter of the impactor was 20 mm. The height between the specimen and the impactor was 1000 mm. The initial drop-velocity of impactor was 4.41 m/s. The initial impact energy given to each specimen was 205 J.

### 3. Results and Discussion

Figure 1 exhibits SEM topography of the pristine p-aramid fiber surface without MWCNT and the p-aramid fiber surface with MWCNT anchored by curing dilute phenolic resin. In comparison, at the same magnification, the pristine p-aramid fiber surface without MWCNT was clear with smoothness. On the other hand, the MWCNT nanoparticles were distributed on the fiber surface, increasing the surface roughness. The anchored MWCNT

may play a bridging role in connecting between the fiber and the matrix of p-AF/VE composite. It was thought that such a MWCNT anchoring effect may contribute to the resistance of the composite to the applied external force and energy.

In our previous study [29], yarn pull-out tests were performed at high speed (800 mm/min) with p-AF containing MWCNT anchored by phenolic resin at various concentrations. Consequently, the highest pull-out force was obtained when 0.05 wt% MWCNT and 0.01 wt% phenolic resin in the MWCNT/phenolic/methanol mixture prepared for anchoring were used, indicating the synergetic effect of MWCNT anchored on the fiber surface by thermally cured phenolic resin. It was found that the friction between the individual fibers consisting of p-AF influenced the increase in the force required for pulling out the single yarn. The result gave us some indications that MWCNT anchoring to the p-aramid fiber surface would play a positive role in improving the thermal, mechanical, and impact properties of p-AF/VE composites.

### 3.1. Thermal Expansion Behavior

Figure 4 displays the thermo-dimensional changes of pristine p-AF/VE and MWCNT-p-AF/VE composites measured by means of TMA. Based on the thermo-dimensional changes in each composite, the coefficients of linear thermal expansion (CLTE) were determined from the slope of each TMA curve in the two temperature ranges of 40~100 °C and 150~250 °C, respectively. The temperature ranges were adapted before and after the temperature showing the drastic dimensional changes between 110~140 °C.

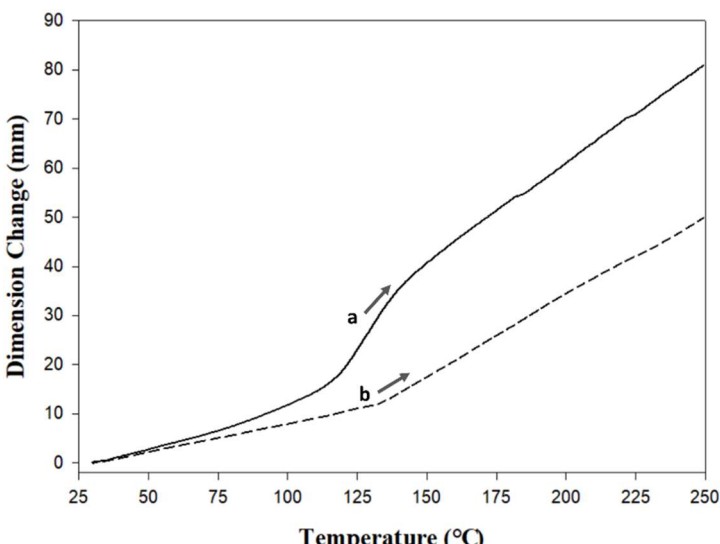

**Figure 4.** Thermo-dimensional changes as a function of temperature measured with (**a**) pristine p-AF/VE and (**b**) MWCNT-p-AF/VE composites.

In the case of pristine p-AF/VE composite, thermal expansion was obviously found in the temperature range of 110~140 °C. It may be attributed to the glass transition behavior of the VE matrix of p-AF/VE composites. Beyond this temperature range, the CLTE value was highly increased from 57.2 μm/m·°C (between 40 and 100 °C) to 134.0 μm/m·°C (between 150 and 250 °C), as listed in Table 1. As similarly found in pristine p-AF/VE composite, the thermal expansion of MWCNT-p-AF/VE composite was apparently observed between 125~135 °C. It was also ascribed to the glass transition behavior of the VE matrix. The thermal expansion of MWCNT-p-AF/VE composite was smaller than that of pristine p-AF/VE composite, showing the CLTE of 38.2 μm/m·°C (between 40 and 100 °C) and 109.4 μm/m·°C (between 150 and 250 °C). This may be explained considering that the MWCNT anchored to the fiber surface of p-AF contributed to increasing the mechanical interlocking and the interfacial adhesion between the VE matrix and the p-AF due to the

increased surface roughness, and consequently contributed to restricting, to some extent, the thermal expansion of the composite.

**Table 1.** Coefficients of linear thermal expansion (CLTE) of p-AF/VE composites determined from two temperature ranges.

| Composite Type | CLTE (µm/m·°C) | |
| --- | --- | --- |
| | **40~100 °C** | **150~250 °C** |
| Pristine p-AF/VE | 57.2 | 134.0 |
| MWCNT-p-AF/VE | 38.2 | 109.4 |

### 3.2. Dynamic Mechanical Properties

Figure 5 displays the variations of the storage modulus and tan δ of pristine p-AF/VE and MWCNT-p-AF/VE composites as a function of temperature measured by means of DMA. The storage modulus of both pristine p-AF/VE and MWCNT-p-AF/VE composites was distinguishably decreased in the glass transition region between 100~160 °C. In this region, the molecular mobility of the VE matrix was increased because of the weakened molecular interaction with temperature. This resulted in the lowering of the storage modulus. The storage modulus of MWCNT-p-AF/VE composite (curve b) was higher than that of pristine p-AF/VE composite (curve a) in the whole temperature range. The storage modulus of MWCNT-p-AF/VE composite was increased by 11% from 8642 to 9596 MPa at 30 °C. This was attributed to the reinforcing effect of p-AF/VE composite with the increased interfacial adhesion between the p-AF and the VE matrix with the assistance of MWCNT anchoring.

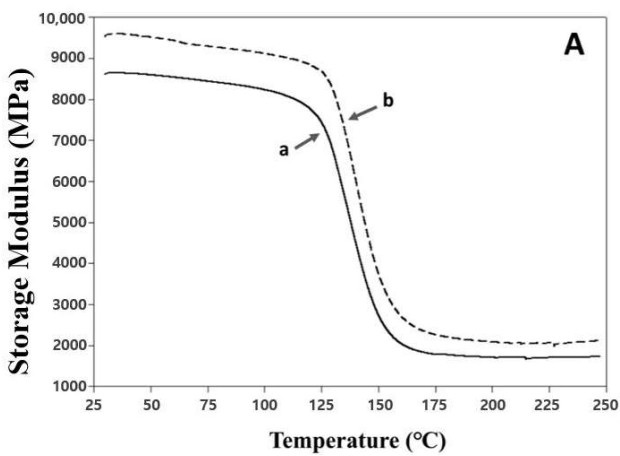 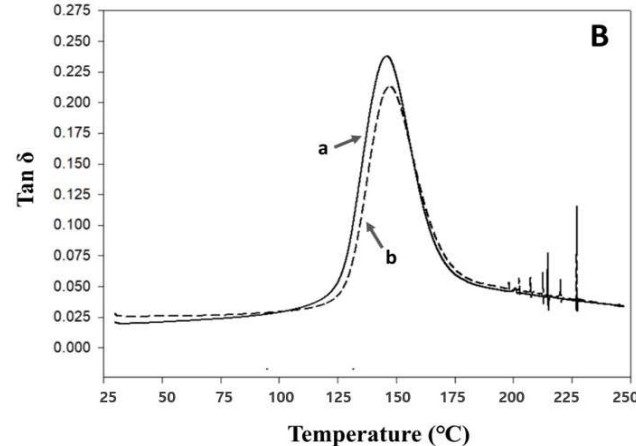

**Figure 5.** Variations of the (**A**) storage modulus and (**B**) tan δ measured with (**a**) pristine p-AF/VE and (**b**) MWCNT-p-AF/VE composites.

The height of tan δ, which is related to the damping property of a material, was decreased as well. It indicates that MWCNT anchoring to the p-aramid fiber surface played a role in dispersing the external load during DMA measurement. The tan δ peak temperature, which is relevant to the glass transition temperature, was slightly shifted to a high temperature by the anchoring effect.

As a result, it may be said that MWCNT anchoring to p-AF contributed to improving the dynamic mechanical properties as well as the thermo-dimensional stability of p-AF/VE composites, forming the mechanical interlocking at the interface between the p-AF and the VE matrix due to the roughened fiber surface.

### 3.3. Heat Deflection Temperature

Heat deflection temperature (HDT) can often be measured by applying a three-point flexural load to the specimen until 0.254 mm deflection occurs in the specimen. Accordingly, the HDT of a polymer composite can be affected by its mechanical resistance, strongly depending on the reinforcement. The MWCNT-p-AF/VE composite was further reinforced by MWCNT anchoring. As described above, it was thought that MWCNT anchoring increased the internal friction between the individual yarns consisting of p-AF during the measurement. In addition, the phenolic resin coated and cured on the fiber surface for anchoring somewhat contributed to increasing the stiffness of p-AF.

As listed in Table 2, the HDT of MWCNT-p-AF/VE composite was about 14 °C higher than that of the pristine p-AF/VE counterpart due to the increased mechanical interlocking between the p-AF and the VE matrix by MWCNT anchoring. This indicates that the loads applied to MWCNT-p-AF/VE composite during measurement were effectively distributed to p-AF through the MWCNT bridges connecting between the individual fibers in the matrix. Therefore, it may be described that MWCNT anchoring played a positive role in resisting the deflection of the p-AF/VE composite by heat.

**Table 2.** A summary of heat deflection temperature, tensile, flexural, and impact properties of pristine p-AF/VE and MWCNT-p-AF/VE composites.

| Properties | P-AF/VE Composites | |
|---|---|---|
| | Pristine P-AF/VE | MMWCNT-p-AF/VE |
| Heat Deflection Temperature (°C) | 241.8 ± 0.3 | 255.4 ± 0.3 |
| Tensile Strength (MPa) | 266 ± 9 | 342 ± 10 |
| Tensile Modulus (GPa) | 7.5 ± 0.4 | 10.0 ± 0.5 |
| Flexural Strength (MPa) | 112 ± 2 | 130 ± 4 |
| Flexural Modulus (GPa) | 15.1 ± 0.5 | 16.2 ± 0.6 |
| Izod Impact Strength (J/m) | 977 ± 9 | 1039 ± 8 |

### 3.4. Mechanical Properties

The stress–strain curves were measured with pristine p-AF/VE and MWCNT-p-AF/VE composites, respectively, as shown in Figure 6. The initial slope and the highest stress obtained with the MWCNT-p-AF/VE composite were higher than those with the pristine p-AF/VE composite. Substantially, the applied load caused the frictional force between the fiber and the matrix of a fiber-reinforced polymer composite material until the specimen was broken during the tensile test. As seen in Figure 1, the fiber surface of MWCNT-p-AF became roughened with the increased surface area by MWCNT anchoring. Accordingly, it was convinced that the anchored MWCNT nanoparticles existing on the fiber surface may contribute to increasing the frictional force between the fiber and the matrix of the resulting composite.

As a result, the tensile strength and modulus required for deforming the MWCNT-p-AF/VE composite were higher than those required for deforming the pristine p-AF/VE composite until the specimen was broken. The tensile strength was increased by about 29%, and the tensile modulus was increased by about 33% due to the MWCNT anchoring effect, as shown in Table 2. It turns out that the tensile loads applied to the MWCNT-p-AF/VE composite were well transferred from fiber to fiber owing to the increased interfacial bonding between the p-aramid fiber and the VE matrix by MWCNT anchoring.

The area under the stress–strain curve is fundamentally related to the equilibrium toughness absorbing the energy given to a material. It was obvious that the area under the curve obtained with the MWCNT-p-AF/VE composite was greater than that with the p-AF/VE composite. Based on that, it was expected that the dynamic impact toughness of the MWCNT-p-AF/VE composite would be higher than that of the p-AF/VE composite.

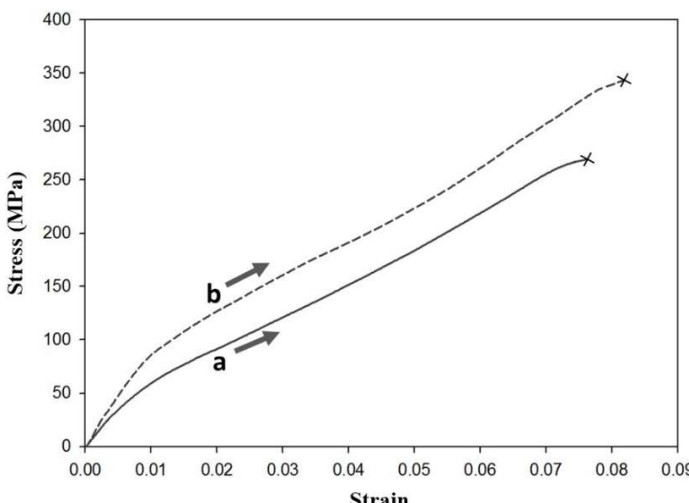

**Figure 6.** Representative stress–strain curves measured with (**a**) pristine p-AF/VE and (**b**) MWCNT-p-AF/VE composites.

Table 2 compares the flexural strength and modulus of p-AF/VE and MWCNT-p-AF/VE composites. The flexural strength and modulus of the MWCNT-p-AF/VE composite was about 16% and 7% higher than that of the pristine p-AF/VE composite, respectively. Upon flexural deformation, a fiber-reinforced composite material is basically affected by both compressive and tensile stresses at the mid-plane of the specimen. The three-point flexural load normally causes compressive stress through the thickness direction of the specimen, which can be mainly governed by the interfacial bonding between the fiber and the matrix of the composite. Meanwhile, the tensile stress can be generated along with the longitudinal direction of the specimen, being influenced by the frictional force between the fiber and the matrix.

As described above, the anchored MWCNT played a role in increasing the frictional force at the interface between the p-AF and the VE matrix, making the composite stronger and stiffer. In our earlier report [29,30], single-yarn pull-out forces at high speed obtained with p-AF only depended not only on the presence and absence of MWCNT anchored to the fiber surface, but also on the concentration of MWCNT and diluted phenolic resin used for anchoring.

It may be insisted that the MWCNT anchoring to the p-aramid fiber surface performed in this work contributed to increasing the fiber surface roughness, the mechanical interlocking between the fiber and the matrix, and the friction at the fiber–matrix interface without fiber damages. As a result, the tensile and flexural properties of p-AF/VE composites were improved as well.

### 3.5. Izod Impact Strength

Table 2 also compares the Izod impact strength of the p-AF/VE and MWCNT-p-AF/VE composites. The impact strength of the pristine p-AF/VE composite was increased by about 6% from 977 to 1039 J/m due to the MWCNT anchoring effect. The result indicates that the applied dynamic impact energy can be efficiently absorbed by the MWCNT-p-AF/VE composite, compared to the pristine p-AF/VE composite. It has been found that the impact resistance increases with increasing interfacial bonding between MWCNT and polymer of a polymer composite containing MWCNT [23]. As described above, in the case of the MWCNT-p-AF/VE composite, MWCNT nanoparticles were physically attached to the p-aramid fiber surfaces by thermally curing them with diluted phenolic resin. They increased the interfacial bonding between the aramid fiber and the VE matrix of the composite. Accordingly, the presence of anchored MWCNT was responsible for the increase in the Izod impact strength of the MWCNT-p-AF/VE composite.

### 3.6. Energy Absorption Behavior by Drop-Weight Impact

Figure 7 displays the results monitored for the variation of the velocity of the impactor as a function of time when the pristine p-AF/VE and MWCNT-p-AF/VE composites were exposed to the drop-weight impact environment, respectively. The initial impactor velocity was 4.41 m/s. The time to reach zero velocity in each composite is relevant to the energy absorption behavior. As shown, the velocity of the impactor was gradually decreased with increasing time, indicating that it was changed from positive to negative after reaching zero velocity. It turns out that the movement of the impactor was changed to the opposite direction relative to the initial direction of the impactor. The time required for reaching zero velocity in the MWCNT-p-AF/VE composite was shorter than that in the pristine p-AF/VE composite.

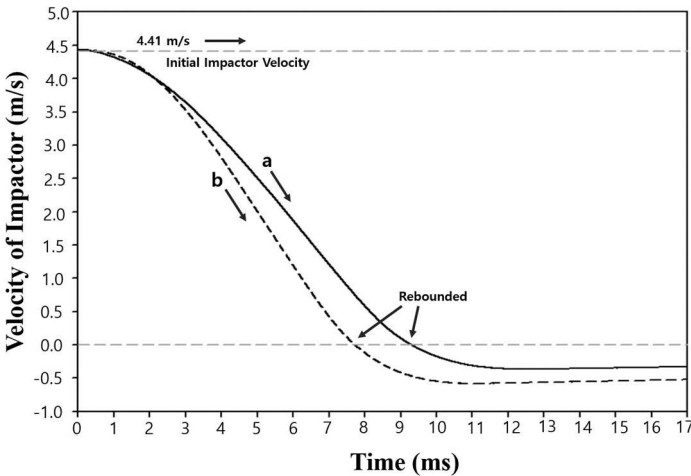

**Figure 7.** Variation of the velocity of impactor as a function of time occurred during drop-weight impact test: (**a**) pristine p-AF/VE and (**b**) MWCNT-p-AF/VE composites.

This can be explained considering that the anchoring of MWCNT to p-AF contributed to releasing the drop-weight impact energy. In addition, the negative velocity value measured with the MWCNT-p-AF/VE composite was lower than that with the pristine p-AF/VE composite, indicating that the drop-weight impactor can be more rebounding or elastically behaving with the MWCNT-p-AF/VE composite than with the pristine p-AF/VE composite. This indicates that the impact resistance of the p-AF/VE composite was increased by the MWCNT anchoring effect, concordantly with the Izod impact strength results and the stress–strain behavior mentioned above.

Figure 8 exhibits the variation of the impact energy absorption as a function of time measured with pristine p-AF/VE and MWCNT-p-AF/VE composites, respectively. The energy absorbed by the pristine p-AF/VE composite corresponded to the initial energy of the impactor given at 9.3 ms, whereas the energy absorbed by the MWCNT-p-AF/VE composite reached the initial impactor energy after an elapse of 7.7 ms upon drop-weight impact. The result indicates that MWCNT anchoring enhanced the energy absorption capability of the composite, rebounding the impactor in the case of the MWCNT-p-AF/VE composite.

Figure 9 shows the variation of the transferred force as a function of time occurring in the pristine p-AF/VE and MWCNT-p-AF/VE composites during the drop-weight impact test, respectively. The time to reach the peak of the transferred force in the MWCNT-p-AF/VE composite was shorter than that in the pristine p-AF/VE composite. The transferred force (17.0 kN) at the peak of the MWCNT-p-AF/VE composite was about 24% higher than that (13.7 kN) of the pristine p-AF/VE composite. The end time of the force transferred in the MWCNT-p-AF/VE composite was about 2 milli-seconds shorter than that in the pristine p-AF/VE composite, similarly to the impactor velocity and the energy absorption. This revealed that the MWCNT anchoring was good to enhance the energy absorption capability and to toughen the p-AF/VE composite, giving rise to the increase in the fiber

surface roughness and the interfacial adhesion between the aramid fiber and the VE matrix of the composite.

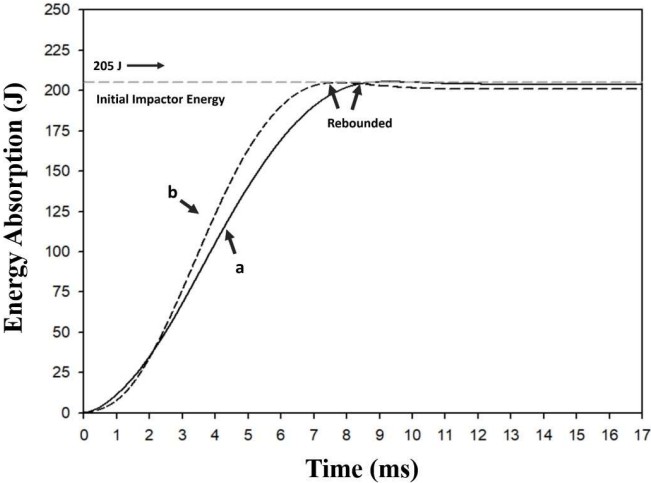

**Figure 8.** Variation of energy absorption as a function of time occurring during drop-weight impact test: (**a**) pristine p-AF/VE and (**b**) MWCNT-p-AF/VE composites.

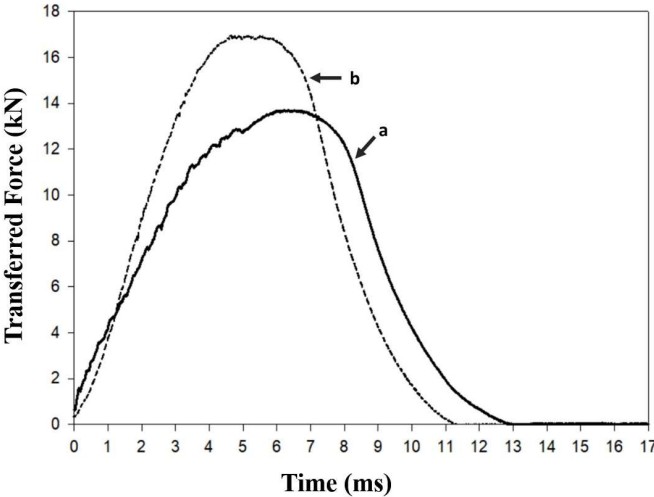

**Figure 9.** Variation of the transferred force as a function of time occurring during drop-weight impact test; (**a**) pristine p-AF/VE and (**b**) MWCNT-p-AF/VE composites.

## 4. Conclusions

The thermo-dimensional, dynamic mechanical, tensile, flexural, and impact properties of the p-AF/VE composite were significantly increased with the assistance of the MWCNT anchoring process, which can physically attach MWCN nanoparticles on the fiber surface by applying the MWCNT/phenolic/methanol mixture to p-AF, and then by thermally curing phenolic resin of very low concentration.

In particular, the drop-weight impact test results revealed that the variations in the velocity of the impactor, the energy absorption, and the transferred force as a function of time monitored for the pristine p-AF/VE and MWCNT-p-AF/VE composites during the impact test agreed with each other. The MWCNT-p-AF/VE composite exhibited a toughness higher than the pristine p-AF/VE composite. The result was consistent with the equilibrium toughness based on the stress–strain behavior and the dynamic toughness based on the Izod impact strength. The improvement on the thermal, mechanical, and impact properties of the MWCNT-p-AF/VE composite can be explained considering that the MWCNT anchoring contributed to increasing the interfacial adhesion between the

p-aramid fiber and the VE matrix, being attributed to the mechanical interlocking by the roughened surface at the fiber–matrix interface.

The present work addresses that the anchoring of MWCNT to p-AF may be desirable to provide an additional benefit to conventional p-AF/polymer composites, further increasing their thermal, mechanical, and impact properties.

**Author Contributions:** Conceptualization, D.C.; writing—original draft preparation, D.C.; writing—review and editing, D.C.; supervision, D.C.; Funding acquisition, D.C.; formal analysis, J.C.; methodology, J.C.; investigation, J.C.; data curation, J.C. All authors have read and agreed to the published version of the manuscript.

**Funding:** This research was supported by the Kumoh National Institute of Technology (2022).

**Institutional Review Board Statement:** Not applicable.

**Informed Consent Statement:** Not applicable.

**Data Availability Statement:** The data presented in this study are available on request from the corresponding author.

**Conflicts of Interest:** The authors declare no conflict of interest.

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
