# Peer review of "Effect of MWCNT Anchoring to Para-Aramid Fiber Surface on the Thermal, Mechanical, and Impact Properties of Para-Aramid Fabric-Reinforced Vinyl Ester Composites"

_jcs, doi:10.3390/jcs7100416_

Round 1

Reviewer 1 Report

Composites structures and materials have been drawing increasing attention recent few years due to high mechanical performance as well as lightweight properties. Author investigated thermal, mechanical and especially the impact performance of para-Ara-3 mid Fabric-Reinforced Vinyl Ester Composites. Although the manuscript was well written, but the reivewer have some comments which need authors addressed, as follows.

1. The introduction needs improvement. Fiber-reinforced composites structures and materials show good potential on solving impact proplems and energyabsorption. The follow paper was recommended.

https://doi.org/10.1016/j.tws.2021.108810

2. Figure 2 is not clear.

3. For the quasi-static condition, why the 4.91 mm/min was used?

4. In Figure 6, there is no need to show the (mm/mm) for the strain.

5. How many repetitive tests?

6. Could authors show some visul columns or results to show the comparision between composite and pure material?

Author Response

Responses to the Reviewer1’s Comments

Composites structures and materials have been drawing increasing attention recent few years due to high mechanical performance as well as lightweight properties. Author investigated thermal, mechanical and especially the impact performance of para-Ara-3 mid Fabric-Reinforced Vinyl Ester Composites. Although the manuscript was well written, but the reivewer have some comments which need authors addressed, as follows.

Comment 1. The introduction needs improvement. Fiber-reinforced composites structures and materials show good potential on solving impact problems and energy absorption. The follow paper was recommended.

https://doi.org/10.1016/j.tws.2021.108810

Response: As recommended, the paper entitled “Progressive collapse behaviors and mechanisms of 3D printed thin-walled composite structures under multi-conditional loading” co-authored by Jin Wang et al. was cited [reference 20] in the Reference part.

The introduction was improved by adding some sentences based on the paper. The revisions were marked in red in the revised manuscript (lines 71 to 74) Reference 20 was added in lines 678-679. Accordingly, the references were renumbered after 20

Comment 2. Figure 2 is not clear.

Response: Figure 2 was revised using new chemical structure with a clear image, as shown in the revised manuscript.

Comment 3. For the quasi-static condition, why the 4.91 mm/min was used?

Response: According to the ASTM D790 standard, in three-point flexural test, the crosshead speed can be adopted, depending on the span-to-depth ratio of the composite specimen. The span-to-depth ratio of the specimen used in this work was 32:1, as described in the manuscript. As recommended in the ASTM standard, the crosshead speed of about 5 mm/min for appropriate three-point flexural test was adopted based on the ratio, considering of our specimen dimensions and testing machine. The crosshead speed of 4.91 mm/min was corrected to 5 mm/min in the revised manuscript for better understanding. The revisions were marked in red (line 283).

Comment 4. In Figure 6, there is no need to show the (mm/mm) for the strain.

Response: The expression (mm/mm) for strain was removed in the revised manuscript.

Comment 5. How many repetitive tests?

Response: The flexural, tensile, and impact tests were repeated 10 times. The test number was described in the revised manuscript. The revisions were marked in red (line 287 to 289).

Comment 6. Could authors show some visual columns or results to show the comparison between composite and pure material?

Response: Based on the reviewer’s comment, the authors understood that the composite and pure material (above-written) are MWCNT-p-AF/VE composite and pristine p-AF/VE composite, respectively. The comparisons between MWCNT-p-AF/VE composite and pristine p-AF/VE composite were given in Table 1 for the coefficients of linear thermal expansion (CLTE) and Table 2 for heat deflection temperature, tensile strength, tensile modulus, flexural strength, flexural modulus, and Izod impact strength. The dynamic mechanical properties and the energy absorption behavior between MWCNT-p-AF/VE composite and pristine p-AF/VE composite were clearly and easily compared in relevant figures (Figures 5, 7, 8, and 9).

Please, kindly consider of this point.

The author would like to appreciate your kind comments indeed.

Reviewer 2 Report

In this paper, various strength tests for the vinyl ester composites reinforced with MWCNT-anchored p-aramid fabrics were conducted. It was found that the strength of aramid fibers is improved by anchoring MWCNT. This paper is very interesting and meaningful for the reinforced fiber field. I recommend that this paper is published after minor revisions according to a following comments.

Comments:

1) L15-19

There are too many items in the text, making it very difficult to understand. It would be better to use the subtitles in "Results and Discussion". For example, “The effect of anchored MWCNT in the p-AF/VE composites on the thermal expansion behavior, dynamic mechanical properties, heat deflection temperature, mechanical properties, Izod impact strength, and energy absorption behavior was investigated.”

2) L300-301

The authors mentioned “The MWCNT nanoparticles were distributed on the fiber surface, increasing the surface roughness.”, but the same magnification image of the p-AF/VE composite without MWCNT should be shown and the surface conditions compared.

3) L157-159

Since “phr” is not a familiar unit, a detailed explanation is required.

Author Response

Responses to the Reviewer2’s Comments

In this paper, various strength tests for the vinyl ester composites reinforced with MWCNT-anchored p-aramid fabrics were conducted. It was found that the strength of aramid fibers is improved by anchoring MWCNT. This paper is very interesting and meaningful for the reinforced fiber field. I recommend that this paper is published after minor revisions according to a following comments.

Comment 1) L15-19

There are too many items in the text, making it very difficult to understand. It would be better to use the subtitles in "Results and Discussion". For example, “The effect of anchored MWCNT in the p-AF/VE composites on the thermal expansion behavior, dynamic mechanical properties, heat deflection temperature, mechanical properties, Izod impact strength, and energy absorption behavior was investigated.”

Response: I totally agree with your suggestion. As indicated in the manuscript, the subtitles in the “Results and Discussion’ part were already used. To avoid some difficulties to understand our findings, the following subtitles were given as below.

3.1. Thermal Expansion Behavior

3.2. Dynamic Mechanical Properties

3.3. Heat Deflection Temperature

3.4. Mechanical Properties

3.5. Izod Impact Strength

3.6. Energy Absorption Behavior by Drop-Weight Impact

Comment 2) L300-301

The authors mentioned “The MWCNT nanoparticles were distributed on the fiber surface, increasing the surface roughness.”, but the same magnification image of the p-AF/VE composite without MWCNT should be shown and the surface conditions compared.

Response: The same magnification image of the P-AF/VE composite without MWCNT was added in the revised manuscript (Figure 1). The relevant sentence was described in the text as well (lines 307 to 310).

Comment 3) L157-159

Since “phr” is not a familiar unit, a detailed explanation is required.

Response: The unit ‘phr’ was revised to ‘pph’ (parts per hundred) in the revised manuscript. The revisions were marked in red (lines 160, 161, and 162).

The author would like to appreciate your kind comments indeed.

Round 2

Reviewer 1 Report

Authors have addressed all my questions and comments.

Reviewer 2 Report

This paper is meaningful for the reinforced fiber field.

The revised paper has been carefully changed according to my comments.

I accept in present form.